# Efficient Channel Feedback Scheme for Multi-User MIMO Hybrid Beamforming Systems

**DOI:** 10.3390/s21165298

**Published:** 2021-08-05

**Authors:** Won-Seok Lee, Hyoung-Kyu Song

**Affiliations:** 1Department of Information and Communication Engineering, Sejong University, Seoul 05006, Korea; scu008@nate.com; 2Department of Convergence Engineering for Intelligent Drone, Sejong University, Seoul 05006, Korea

**Keywords:** channel feedback, compressive sensing, hybrid beamforming, MU-MIMO, mmWave

## Abstract

This paper proposes an efficient channel information feedback scheme to reduce the feedback overhead of multi-user multiple-input multiple-output (MU-MIMO) hybrid beamforming systems. As massive machine type communication (mMTC) was considered in the deployments of 5G, a transmitter of the hybrid beamforming system should communicate with multiple devices at the same time. To communicate with multiple devices in the same time and frequency slot, high-dimensional channel information should be used to control interferences between the receivers. Therefore, the feedback overhead for the channels of the devices is impractically high. To reduce the overhead, this paper uses common sparsity of channel and nonlinear quantization. To find a common sparse part of a wide frequency band, the proposed system uses minimum mean squared error orthogonal matching pursuit (MMSE-OMP). After the search of the common sparse basis, sparse vectors of subcarriers are searched by using the basis. The sparse vectors are quantized by a nonlinear codebook that is generated by conditional random vector quantization (RVQ). For the conditional RVQ, the Linde–Buzo–Gray (LBG) algorithm is used in conditional vector space. Typically, elements of sparse vectors are sorted according to magnitude by the OMP algorithm. The proposed quantization scheme considers the property for the conditional RVQ. For feedback, indices of the common sparse basis and the quantized sparse vectors are delivered and the channel is recovered at a transmitter for precoding of MU-MIMO. The simulation results show that the proposed scheme achieves lower MMSE for the recovered channel than that of the linear quantization scheme. Furthermore, the transmitter can adopt analog and digital precoding matrix freely by the recovered channel and achieve higher sum rate than that of conventional codebook-based MU-MIMO precoding schemes.

## 1. Introduction

In conventional wireless communication systems, beamforming techniques are optional as the signal to noise power ratio (SNR) of received signals is enough for nearly error-free communication with modern channel coding techniques. Furthermore, techniques to increase the usable frequency bandwidth and the number of independent streams were more efficient for the data rate than techniques for high received SNR. However, the resources of the frequency band have been exhausted with the massive growth of wireless devices such as smartphones, IoT machines, and the wireless infrastructure of cities. The mMTC deployment of 5G was considered for the case that massive wireless connections exist. Furthermore, measurements have shown that wireless channels cannot provide independent spatial paths proportional to the number of antennas [1,2,3]. To accommodate the traffic of massive wireless devices, many wireless systems consider the use of millimeter-wave (mmWave) frequency bands over 30 GHz. The one feature of mmWave frequency bands is high path loss. Due to the high path loss, beamforming becomes essential to wireless systems that use the millimeter frequency bands.

For beamforming of mmWave frequency bands, massive MIMO systems have been studied [4,5]. In massive MIMO systems, the antenna number of a transmitter is larger than four times the total antenna number of receivers. The systems can provide nearly optimal SNR gain with zero-forcing (ZF) beamforming. However, the same number of RF chains with the antenna number is an infeasible constraint. To ease the constraint, hybrid beamforming where analog and digital beamforming are combined appeared [4,5,6,7,8,9]. Analog beamforming uses only phase shifters to make beamforming gain by combining the multiple same signals coherently at the desired direction. In hybrid beamforming, the gain of analog beamforming complements the loss of digital beamforming gain that is caused by reduced RF chains. As analog beamforming is conducted by phase shifters, the signal processing is modeled by a matrix that the elements are complex values of unit modulus. The constraint of the analog beamforming matrix makes the joint optimization of the beamforming matrices a non-convex problem. There have been many studies concerning the non-convex optimization in flat and selective fading channel [5,7,8,9]. Most of those studies have used alternate optimization assuming perfect channel state information at a transmitter (CSIT). However, in a real environment, it is difficult that a transmitter acquires the estimated channel matrices due to feedback overhead and almost wireless systems use a codebook of precoding matrices for feedback [10,11,12,13]. In the real systems, wireless transceivers that adopt hybrid beamforming use a protocol for beam management to determine analog beams [14,15,16]. The protocols determine the best beam pair for each link among beams of a predefined codebook and the selected indices of the pair are delivered to the transmitter as feedback information. Although the protocols provide realistic ways for hybrid beamforming, the feedback of precoding matrices does not allow the joint optimization of the hybrid beamforming structure.

Another promising technique to accommodate the growth of wireless devices is multi-user MIMO (MU-MIMO) [17,18]. Wireless systems that use MU-MIMO precoding can transmit independent data streams to multiple receivers stably in the same time-frequency resource block. By MU-MIMO technique, wireless systems can achieve better throughput than that of single-user MIMO (SU-MIMO) technique. However, for stable transmission, suppression of interference between receivers is essential. If high-dimensional CSIT is satisfied, the inter-user interference (IUI) is suppressed efficiently by MU-MIMO precoding. The feedback of precoding matrices is inefficient for suppressing dynamic IUI. Imperfect precoding severely degrades the performance of MU-MIMO systems [12,13].

To realize beamforming and MU-MIMO systems sufficiently, the improvement of channel estimation and feedback techniques is important. For channel estimation of hybrid beamforming systems, many schemes have been proposed in frequency-division duplex (FDD) and time-division duplex (TDD) environments [19,20,21,22,23,24,25]. In FDD systems, uplink and downlink channels use different bands. Therefore, for CSIT, the receiver must allocate some resources of an uplink channel to deliver channel information. The recent trend that increases the number of antennas has intensified the feedback overhead. In TDD systems, uplink and downlink channels use the same frequency band. By exploiting the reciprocity, CSIT for downlink can be achieved from pilot signals of the uplink channel. As CSIT can be achieved from the uplink channel, the overhead for CSIT is proportional to the number of receivers. However, channel estimation by using the reciprocity is not always possible due to various configurations of transmission and reception modes. Therefore, to cope with the complex configuration, TDD systems also need the feedback of channels for CSIT.

In massive MIMO systems, generally, the antenna number of a transmitter is much larger than the total antenna number of receivers. By the rate of the antenna numbers, the least-square (LS) method does not provide reasonable performance for channel estimation. In the mmWave frequency bands, the wireless channel shows sparsity in the angle domain as most signals of multiple paths are absorbed and removed easily by the surrounding environment [1,3]. In the case of MIMO channel, the sparsity is also observed since the received signals of multiple antennas propagate through similar paths [3]. By exploiting the sparsity, compressed sensing (CS) can provide reasonable performance for channel estimation although the rate of antenna numbers is high [19,20,21,22,26,27,28]. The schemes can be classified into three groups. The schemes of the first group have focused on the design of beam sweeping vectors to acquire qualified measurements for CS [19,26], and the schemes of the second group provided methods to reduce the computational load of the conventional OMP operation [21,22,26,28]. The schemes of the last group utilized structure of channel basis to improve the accuracy of channel estimation [20,27,28]. The topics are important subjects of CS-based channel estimation. However, the schemes did not provide a practical feedback method of the estimated channel. Another category of CS-based channel estimation is hierarchical search (HS). HS schemes have focused on reduction of overhead for the measurements [23,24,25]. To reduce the overhead, the proposed schemes in [23,24,25] optimized search regions for the measurements by the feedback of search results. However, the schemes need multiple reports for channel estimation within a coherent time block. The closed-loop can be a high overhead for MU-MIMO systems due to protocol for multiple receivers and these schemes also did not provide a feedback method for the estimated channel.

Some estimation methods adopted machine learning (ML) schemes [25,29,30,31]. The ML-based methods also focused on estimation performance and shown better performance than that of the CS-based methods. Among the proposed systems, the authors of [30,31] considered feedback schemes of receivers. The feedback schemes designed pilots of a transmitter jointly with the structure of receivers. However, the joint designs only consider low-dimensional baseband channels.

## 2. Contributions and Notations

In this paper, a feedback scheme is proposed to reduce the feedback overhead of high-dimensional RF channels and improve the quality of the quantized estimated channel. The estimation and recovery of channels are performed by CS.

To reduce the feedback overhead, only sparse vectors are delivered for each subcarrier with a common sparse basis of all subcarriers. The matrix for the common sparse basis is delivered as indices of a pre-shared codebook. In this paper, the elements of the codebook are assumed as column vectors of the discrete Fourier transform (DFT) matrix. The sparse vectors are quantized by a low-dimensional codebook. The dimension of the codebook is only proportional to the estimated sparsity of the wireless channel. As the quantization is performed on only the sparse vectors, the reduction and quality improvement of feedback information is achieved.The proposed scheme also uses the nonlinear codebook to improve the quality of CSIT. The sparse vectors of the common basis are calculated by the OMP method. After the calculation, elements of the sparse vectors are sorted by order of magnitudes. The nonlinear codebook is generated by considering the property of the sparse vectors. The quantized vectors by the nonlinear codebook can achieve lower error than that of a linear codebook with the same codebook size.By the feedback of the proposed scheme, a transmitter can recover more accurate high-dimensional channels than recovered channel by linearly quantized pilots with the same codebook size. The recovered high-dimensional channels allow the transmitter to optimize jointly the analog and digital beamformer. Furthermore, a higher sum rate can be achieved than that of feedback using a codebook of precoding matrices.

This paper uses the lower-case letters for scalars, lower-case bold letters for vectors, and upper-case bold letters for matrices. Further, $C^{a\times b}$ for a-by-b matrix of complex elements, || || for norm operation, and $\mathrm{diag}\left(\right)$ for diagonalization of a vector are used. For transpose and conjugate transpose of matrix A, $A^{T}$ and $A^{H}$ are used. ${\left[A\right]}_{a,b}$ means the element of A at the *a*-th low and the *b*-th column. ${\left[A\right]}_{:,b}$ means the *b*-th column vector of A.

## 3. System Model

In this paper, a multi-user hybrid beamforming system is considered. In the system, it is assumed that beam sweeping is used to acquire CSI for beamforming. Figure 1 shows the beam sweeping operation of the multi-user hybrid beamforming system. The system includes a transmitter and multiple receivers. The transmitter is consisted of digital beamformer following OFDM modulator and analog beamformer. $N_{s}$, $N_{RF}$, and $N_{t}$ are the number of independent data streams, RF chains, and antennas for transmission, respectively. In the structure, $N_{RF}$ is set to hold a condition that $N_{s}⩽N_{RF}<N_{t}$. To consider MU-MIMO transmission, the same number of receivers are assumed with $N_{s}$. Each receiver consists of an analog combiner with $N_{r}$ antennas following a RF chain. At the same time as the beam sweeping operation the transmitter sweeps channel with $M_{t}$ beams. The receivers also sweep the channel with $M_{r}$ beams. The measurements by the beam sweeping are as follows:(1)$$R_{u}\left[k\right]=Q^{H}H_{u}\left[k\right]P+Q^{H}Z_{u}\left[k\right],$$
where
(2)$$Q=\left[\begin{array}{cccc} q_{1} & q_{2} & \cdots & q_{M_{r}} \end{array}\right],$$
(3)$$P=\left[\begin{array}{cccc} p_{1} & p_{2} & \cdots & p_{M_{t}} \end{array}\right].$$
$q_{n}\in C^{N_{r}}$ and $p_{n}\in C^{N_{t}}$ are vectors for the beam sweeping at the receivers and the transmitter. $H_{u}\left[k\right]\in C^{N_{r}\times N_{t}}$ is a channel matrix of frequency domain between the transmitter and the *u*-th receiver. *k* is an index of a OFDM subcarrier. $Z_{u}\left[k\right]\in C^{N_{r}\times M_{t}}$ is a noise matrix at the *u*-th receiver. The elements of $R_{u}\left[k\right]\in C^{M_{r}\times M_{t}}$ represent the measurements by the beam pairs. After the beam sweeping operation of the transmitter and the receivers, the best beam pairs of each receiver can be selected from the measurements and used for analog beamforming. Furthermore, the digital beamformer can use pre-shared precoding matrices based on baseband channels of the selected beam pairs.

For the channel model, a three-dimensional statistical channel model is considered. In the time domain, the channel matrix $H_{l_{u}}\in C^{N_{r}\times N_{t}}$ of the model is calculated as follows:(4)$$H_{l_{u}}=\sqrt{\frac{N_{t}N_{r}}{L_{u}C_{u}}}A_{r,l_{u}}D_{l_{u}}A_{t,l_{u}}^{H},$$
where
(5)$$\begin{array}{c} A_{r,l_{u}}\\ =\left[\begin{array}{ccc} a_{r}\left(\phi_{l_{u},1}^{r},\theta_{l_{u},1}^{r}\right) & \cdots & a_{r}\left(\phi_{l_{u},C_{u}}^{r},\theta_{l_{u},C_{u}}^{r}\right) \end{array}\right], \end{array}$$
(6)$$\begin{array}{c} A_{t,l_{u}}\\ =\left[\begin{array}{ccc} a_{t}\left(\phi_{l_{u},1}^{t},\theta_{l_{u},1}^{t}\right) & \cdots & a_{t}\left(\phi_{l_{u},C_{u}}^{t},\theta_{l_{u},C_{u}}^{t}\right) \end{array}\right], \end{array}$$
(7)$$D_{l_{u}}=\mathrm{diag}\left(\left[\begin{array}{cccc} {\alpha_{l_{u},}}_{1} & \alpha_{l_{u},2} & \cdots & \alpha_{l_{u}C_{u}} \end{array}\right]\right).$$
$A_{r,l_{u}}\in C^{N_{r}\times C_{u}}$ and $A_{t,l_{u}}\in C^{C_{u}\times N_{t}}$ are response matrices at the transmitter and the receiver sides. $D_{l_{u}}\in C^{C_{u}\times C_{u}}$ is a diagonal matrix that reflects distortion of amplitude and phase for transmitted signals. $L_{u}$, and $C_{u}$ means number of propagation paths in channel of the *u*-th receiver and the number of scatterers in the $l_{u}$-th path. $\alpha_{l_{u},c_{u}}∼\mathrm{CN}$$\left(0,g_{l_{u},c_{u}}\right)$ and $g_{l_{u},c_{u}}$ are complex and real gains of the $c_{u}$-th scatterer in the $l_{u}$-th path. $a_{r}\left(\phi_{l_{u},c_{u}}^{r},\theta_{l_{u},c_{u}}^{r}\right)$ and $a_{t}\left(\phi_{l_{u},c_{u}}^{t},\theta_{l_{u},c_{u}}^{t}\right)$ are response vectors at the transmitter and the receiver sides. $\phi_{l_{u},c_{u}}$ and $\theta_{l_{u},c_{u}}$ are azimuth and elevation through the $c_{u}$-th scatterer of the $l_{u}$-th path, respectively. The response vector for azimuth ϕ and elevation θ is as follows: (8)$$\begin{array}{c} a_{a}\left(\phi ,\theta \right)\\ =\frac{1}{\sqrt{N_{a}}}{\left[\begin{array}{ccccc} 1 & \cdots & e^{j\frac{2\pi }{\lambda }d_{0}^{T}d_{p}} & \cdots & e^{j\frac{2\pi }{\lambda }d_{0}^{T}d_{N_{a}}} \end{array}\right]}^{T}, \end{array}$$
(9)$$d_{0}={\left[\begin{array}{ccc} \cos\phi \sin\theta & \sin\phi \sin\theta & \cos\theta \end{array}\right]}^{T},$$
(10)$$d_{p}={\left[\begin{array}{ccc} d_{x} & d_{y} & d_{z} \end{array}\right]}^{T},$$
where $d_{0}$ and $d_{p}$ are vectors for three-dimensional direction and position of the *p*-th antenna element, respectively. λ is wavelength of the transmitted signals. After demodulation, the channel matrix should be considered in frequency domain. The channel matrix of frequency domain is calculated from the matrix of time domain as follows,:(11)$$H_{u}\left[k\right]=\sum_{l_{u}=1}^{L_{u}}H_{l_{u}}e^{j2\pi kl_{u}/L_{u}}.$$

## 4. Hybrid Beamforming Based on CSI

In multi-user hybrid beamforming systems, a received signal of each receiver is expressed as follows:(12)$$y\left[k\right]=\frac{P}{\gamma \left[k\right]}w_{u}^{H}H_{u}\left[k\right]\mathrm{FV}\left[k\right]s_{u}\left[k\right],$$
where
(13)$$F=\left[\begin{array}{ccccc} f_{1} & \cdots & f_{u} & \cdots & f_{N_{Rf}} \end{array}\right],$$
(14)$$V\left[k\right]=\left[\begin{array}{ccccc} v_{1}\left[k\right] & \cdots & v_{u}\left[k\right] & \cdots & v_{N_{s}}\left[k\right] \end{array}\right],$$
(15)$$\gamma \left[k\right]=\frac{1}{{∥\mathrm{FV}\left[k\right]∥}^{2}}.$$
$s_{u}\left[k\right]$ is a signal for the *u*-the receiver and $E\left\{s_{u}\left[k\right]s_{u}^{H}\left[k\right]\right\}=1$. *P* and $\gamma \left[k\right]$ are average power for transmission and a constant for normalization. $\gamma \left[k\right]$ is also viewed as beamforming gain in massive MIMO systems. $w_{u}\in C^{N_{r}}$ is a analog combiner of the *u*-th receiver for Rx beamforming. $f_{u}\in C^{N_{t}}$ and $v_{u}\left[k\right]\in C^{N_{RF}}$ are analog and digital Tx beamforming vectors for the *u*-th receiver. For $y\left[k\right]$, spectral efficiency is calculated using signal to interference plus noise ratio (SINR) as follows [9]:(16)$$S_{u}\left[k\right]=\log_{2}\left(1+\frac{P}{\gamma \left[k\right]}{\tilde{h}}_{u}^{H}\left[k\right]v_{u}\left[k\right]v_{u}^{H}\left[k\right]{\tilde{h}}_{u}\left[k\right]\Phi^{-1}\left[k\right]\right),$$
where
(17)$$\begin{array}{c} \Phi \left[k\right]\\ =\frac{P}{\gamma \left[k\right]}\sum_{j=1,j\neq u}^{N_{s}}{\tilde{h}}_{u}^{H}\left[k\right]v_{j}\left[k\right]v_{j}^{H}\left[k\right]{\tilde{h}}_{u}\left[k\right]+\sigma^{2}w_{u}^{H}w_{u}, \end{array}$$
(18)$${\tilde{h}}_{u}^{H}\left[k\right]=w_{u}^{H}H_{u}\left[k\right]F.$$

In (Equation 16), $\frac{P}{\gamma \left[k\right]}{\tilde{h}}_{u}^{H}\left[k\right]v_{u}\left[k\right]v_{u}^{H}\left[k\right]{\tilde{h}}_{u}\left[k\right]$ is the part that belongs to the desired signal and $\Phi \left[k\right]$ is the part that includes interference and noise. ${\tilde{h}}_{u}\left[k\right]\in C^{N_{RF}}$ is baseband channel of the *u*-th receiver. The transmitter should determine F and $v_{u}\left[k\right]$ for each receiver to maximize the sum rate as follows:(19)$$F^{†},v_{1}^{†}\left[k\right],\cdots ,v_{N_{s}}^{†}\left[k\right]=\underset{F,v_{1}\left[k\right],\cdots ,v_{N_{s}}\left[k\right]}{\arg\max}\sum_{u=1}^{N_{s}}S_{u}\left[k\right].$$

However, in most modern wireless systems, searching for the optimal matrices is impossible as the transmitter can not acquire $H_{u}\left[k\right]$ or ${\tilde{h}}_{u}$. In the systems, the transmitter only acquires $f_{u}$ and the quantized version of $w_{u}^{H}H_{u}\left[k\right]f_{u}$ from feedback of the each receiver. The receivers acquire the information from received baseband pilots that are transmitted by beam sweeping operation. For multi-user transmission, the transmitter determines analog and digital beamforming matrices separately based on the feedback information.

If a transmitter can acquire $H_{u}\left[k\right]$ or ${\tilde{h}}_{u}$, the transmitter can determine the analog and digital beamforming vectors to maximize beamforming gain $\gamma \left[k\right]$ and suppress IUI of $\Phi \left[k\right]$. The example is a conventional beam-steering method. The beam-steering method is a hybrid beamforming method that determines analog beamformer as the reported best beam and digital beamformer as the ZF matrix for ${\left[\begin{array}{cccc} {\tilde{h}}_{1}\left[k\right] & {\tilde{h}}_{2}\left[k\right] & \cdots & {\tilde{h}}_{N_{s}}\left[k\right] \end{array}\right.}^{H}$.

## 5. Quantized Channel Feedback Based on Non-Linear Quantization of Sparse Vectors

Originally, CS was invented to recover an original sparse signal from small measurements of the signal. Typically, most of natural signals show high sparsity. Thus, CS can achieve high performance when compressing and recovering the signals. In the case of the mmWave channel, sparsity is observed in angle domain. After beam sweeping operation and OFDM demodulation, formulation of CS for compressive channel estimation is as follows:(20)$$\min_{e_{u}\left[k\right]}{∥e_{u}\left[k\right]∥}_{0}\;\mathrm{subject}\;\mathrm{to}∥r_{u}\left[k\right]-P^{T}\Psi e_{u}\left[k\right]∥⩽\varepsilon .$$
$r_{u}\left[k\right]$, Ψ, and $e_{u}\left[k\right]$ are the measurements by Tx beam sweeping, a sparse basis matrix, and a sparse vector. ε means magnitude of noise. When Rx beam sweeping is performed, $r_{u}\left[k\right]$ is determined as follows:(21)$$r_{u}\left[k\right]=\left(P^{T}H_{u}^{H}\left[k\right]+Z_{u}^{H}\left[k\right]\right)q_{\beta },$$
where
(22)$$\beta =\underset{i}{\arg\max}\sum_{k=1}^{K}∥\left(P^{T}H_{u}^{H}\left[k\right]+Z_{u}^{H}\left[k\right]\right)q_{i}∥.$$
$q_{\beta }\in C^{N_{r}}$ is the selected vector among the column vectors of Q. *K* means the number of OFDM subcarriers. $r_{u}\left[k\right]$ of the receivers that do not support Rx beam sweeping is just a measurement vector by only Tx beam sweeping. For P and Q, various matrices can be adopted [19,20,21,22]. In this paper, discrete Fourier transform (DFT) matrix is used for Tx and Rx beam sweeping. Typically, random matrices for measurements achieve high recovery performance at compressive sensing due to incoherence with the sparse basis. However, the random matrices spread power and significantly decrease SNR of the measurements. When DFT matrices are used for the measurements, P and Q are calculated as follows: (23)$$\begin{array}{c} P\\ =\left[\begin{array}{cccc} p\left(-\frac{1}{2}\right) & p\left(-\frac{1}{2}+\frac{1}{M_{t}}\right) & \cdots & p\left(\frac{1}{2}-\frac{1}{M_{t}}\right) \end{array}\right], \end{array}$$
(24)$$\begin{array}{c} Q\\ =\left[\begin{array}{cccc} q\left(-\frac{1}{2}\right) & q\left(-\frac{1}{2}+\frac{1}{M_{r}}\right) & \cdots & q\left(\frac{1}{2}-\frac{1}{M_{r}}\right) \end{array}\right], \end{array}$$
where
(25)$$p\left(m\right)={\left[\begin{array}{cccc} 1 & e^{-j2\pi m} & \cdots & e^{-j2\pi m\left(N_{t}-1\right)} \end{array}\right]}^{T},$$
(26)$$q\left(m\right)={\left[\begin{array}{cccc} 1 & e^{-j2\pi m} & \cdots & e^{-j2\pi m\left(N_{r}-1\right)} \end{array}\right]}^{T}.$$

In the matrices, $M_{t}$ and $M_{r}$ must be equal to $N_{t}$ and $N_{r}$, respectively. In other words, the number of time slots for beam sweeping is proportional to the number of antennas. This can be a large overhead for initial access. Furthermore, DFT matrix can be used for only linear array antenna. However, the design of the sweeping matrices is not a focus of this paper and DFT matrices can provide qualified measurements for compressive channel estimation.

There are several methods to find the solution of (Equation 20). The non-convex minimization with the zero-norm condition for the sparse vector provides the best solution but the complexity is impractically high. Among the methods, OMP is the simplest method. OMP is a greed algorithm that finds the best basis matrix and the correspond sparse coefficients iteratively. In this paper, OMP is used to find basis matrix and sparse vectors of channel. The formulation for OMP is as follows [19]:(27)$$\begin{array}{c} \min_{e\left[k\right]}∥r_{u}\left[k\right]-P^{T}\Psi e_{u}\left[k\right]∥\\ \mathrm{subject}\;\mathrm{to}\;{∥e_{u}\left[k\right]∥}_{0}⩽L. \end{array}$$

In (Equation 27), the condition for the sparse vector is relaxed for lower complexity. In the OMP algorithm, the elements of the sparse vector are found by iterative search from the largest element. As the sparse vector has few non-zero elements, the sparse vector can be used for feedback information to reduce the feedback overhead.

### 5.1. Channel Feedback Using Common Sparse Basis

Measurements by $q_{\beta }$ can be represented as follows:(28)$$\begin{array}{c} r_{u}\left[k\right]\\ =P^{T}\sum_{l_{u}=1}^{L_{u}}\left(A_{t,l_{u}}D_{l_{u}}^{T}A_{r,l_{u}}^{T}e^{-j2\pi kl_{u}/L_{u}}\right)q_{\beta }+Z_{u}^{H}\left[k\right]q_{\beta }\\ =P^{T}A_{u}b_{u}\left[k\right]+Z_{u}^{H}\left[k\right]q_{\beta }, \end{array}$$
where
(29)$$A_{u}=\left[\begin{array}{cccc} A_{t,1} & A_{t,2} & \cdots & A_{t,L_{u}} \end{array}\right],$$
(30)$$b_{u}\left[k\right]=\left[\begin{array}{c} D_{1}^{T}A_{r,1}^{T}q_{\beta }e^{-j2\pi k/L_{u}}\\ D_{2}^{T}A_{r,2}^{T}q_{\beta }e^{-j2\pi k2/L_{u}}\\ \vdots \\ D_{L_{u}}^{T}A_{r,L_{u}}^{T}q_{\beta }e^{-j2\pi k} \end{array}\right].$$
$A_{u}\in C^{N_{t}\times C_{u}L_{u}}$ and $b_{u}\left[k\right]\in C^{C_{u}L_{u}}$ are a basis matrix and a vector of coefficients for the basis. The first term of (Equation 28) shows the similar structure with $P^{T}\Psi e_{u}\left[k\right]$. In other words, Ψ and $e_{u}\left[k\right]$ can be mapped to $A_{u}$ and $b_{u}\left[k\right]$. Furthermore, note that $A_{u}$ is independent of *k*. Therefore, the channel coefficients of the every subcarrier can be recovered using the common $A_{u}$. The only feedback information of the each subcarrier is $b_{u}\left[k\right]$ except $A_{u}$.

In channel recovery with the common basis, the quality of the selected basis is a significant factor of recovery performance. Incorrect selection severely degrades the performance of channel recovery, and this is caused by noise. To search for the common basis efficiently, MMSE-OMP is used. As MMSE-OMP considers the effect of the noise, the quality of the recovered channel at a low SNR environment can be improved. In the process, the common basis is searched from a conjugated DFT matrix. Algorithm 1 shows the search process. In the algorithm, Γ, $\hat{\Psi }$, and $\hat{b}\left[k\right]$ represent a constant for normalization, a matrix of the selected basis, and a estimated sparse vector of the *k*-th subcarrier, respectively. As a result of Algorithm 1, $\hat{\Psi }$ and ${\hat{b}}_{u}\left[k\right]$ are found from the received measurements. As $\hat{\Psi }$ is a submatrix of the DFT matrix for Tx beam sweeping and independent on *k*, uplink resources for CSI feedback can be significantly saved. Furthermore, ${\hat{b}}_{u}\left[k\right]$ can be quantized for the feedback by using a codebook that the dimension of the codebook is only dependent on the sparsity of wireless channel. Due to the sparsity of the mmWave wireless channels, the codebook can be searched more efficiently than a codebook for quantizing high-dimensional matrices.
**Algorithm 1** Search of Common Sparse Basis and Corresponding Sparse Vectors$\mathrm{Input}:\;r_{u}\left[k\right]\in C^{M_{t}},P\in C^{N_{t}\times M_{t}},\sigma_{u}$$\mathrm{Require}:\;\hat{\Psi }\leftarrow \left[\right]$$\mathrm{Step}1:\;\mathrm{Constitute}\;\mathrm{basis}\;\mathrm{pool}\;\tilde{\Psi }\;\mathrm{as}\;{\left\{P^{T}\right\}}^{H}$$\mathrm{Step}2:\;\mathrm{Select}\;L\;\mathrm{columns}\;\mathrm{from}\;\tilde{\Psi }$**for**l=1→L**do**  $i_{sel}\leftarrow \underset{i}{\arg\max}{\left\{P^{H}{\left[\tilde{\Psi }\right]}_{:,i}+\sigma_{u}^{2}I_{M_{t}}\right\}}^{H}\sum_{k=1}^{K}r_{u}\left[k\right]$  $\hat{\Psi }\leftarrow \left[\begin{array}{cc} \hat{\Psi } & {\left[\tilde{\Psi }\right]}_{:,i_{sel}} \end{array}\right]$**end for**$\mathrm{Step}3:\;\mathrm{Find}\;\mathrm{the}\;\mathrm{sparse}\;\mathrm{vectors}\;\mathrm{with}\;\hat{\Psi }$**for**k=1→K**do**  ${\hat{b}}_{u}\left[k\right]\leftarrow \frac{1}{\Gamma }{\left\{P^{H}\hat{\Psi }+\sigma_{u}^{2}I_{M_{t}}\right\}}^{H}r_{u}\left[k\right]$**end for**$\mathrm{Output}:\;\hat{\Psi },{\hat{b}}_{u}\left[k\right]$

### 5.2. Nonlinear Codebook Generation

To recover the channel matrix at the transmitter, the matrix for the common basis and the sparse vectors must be delivered to the transmitter. In the case of the basis matrix, the selected columns by Algorithm 1 can be delivered in the form of indices. For the sparse vectors, quantization is an inevitable process. The sparse vectors of OMP show a property that the elements of the vector are sorted according to the magnitude. By considering the order of the magnitude, quantization can be performed more effectively than simple linear quantization. It is well known that the Grassmannian manifold provides an optimal codebook for complex unit vectors. However, it is difficult to use the Grassmannian manifold method for conditional vector space. In this paper, conditional RVQ is used for the quantization of conditional vector space and the nonlinear codebook is generated by the LBG algorithm in conditional vector space [32]. Figure 2 shows an example of a codebook for conditional RVQ generated by the LBG algorithm.

In Figure 2, green and blue dots mean unselected and selected vectors by a specific condition, respectively. The codebook is generated from the selected vectors. The red dots mean the vectors of the codebook. The generated codebook of Figure 2 is calculated in the conditional vector space that the angle between the purple vector and the arbitrary green vector is lower than $\frac{\pi }{3}$.

Algorithm 2 details generation of a codebook for the sparse vectors. In Algorithm 2, the condition is as follows:(31)$$\left|{\left[T\right]}_{1,i}\right|>\left|{\left[T\right]}_{2,i}\right|>\cdots \left|{\left[T\right]}_{L,i}\right|,$$
where the column vectors of $T\in C^{L\times T}$ constitute space that sparse vectors of wireless channel can be observed. By the condition, the vectors of the space are filtered. Then, the filtered column vectors of T are grouped into $G_{1},G_{2},\cdots G_{2^{B}}$ according to distances with the vectors of the codebook. *B* means the size of the codebook. The vectors of the codebook are calculated by averaging the column vectors of the each group. mean() calculates an average vector of the groups. The calculation of the codebook is repeated until convergence is observed. η of Algorithm 2 is a small number close to zero. The output of Algorithm 2 is the codebook B and B is used to quantize the estimated sparse vector $\hat{b}\left[k\right]$ as follows:(32)$${\left[B\right]}_{:,i_{sel}}=\underset{{\left[B\right]}_{:,i}}{\arg\min}∥\left\{\hat{b}\left[k\right]/∥\hat{b}\left[k\right]∥\right\}-{\left[B\right]}_{:,i}∥.$$

In (Equation 32), only phase of $\hat{b}\left[k\right]$ is quantized. The magnitude can be quantized more efficiently than the phase as the dimension is one. Therefore, this paper considers the case that magnitude and phase are quantized separately.
**Algorithm 2** Codebook Generation for Conditional RVQInput:L,B,T$\mathrm{Require}:\;G_{1},G_{2}\cdots ,G_{2^{B}}\leftarrow \left[\right],\Delta \in C^{2^{B}}\leftarrow 0,$$T\in C^{L\times T}∼\mathrm{CN}\left(0,I_{L}\right),B_{n}\in C^{L\times 2^{B}}\leftarrow 0,\;n\in N$Step1:NormalizemagnitudeofcolumnvectorsofTStep2:SelectcolumnvectorsofT**for**i=1→T **do**  **if**
$\mathrm{not}\;\left\{\left|{\left[T\right]}_{1,i}\right|>\left|{\left[T\right]}_{2,i}\right|>\cdots \left|{\left[T\right]}_{L,i}\right|\right\}$
**then**   ${\left[T\right]}_{1,i}\leftarrow \left[\right]$  **end if****end for**Step3:GroupcolumnvectorsofTn←1**while**diff<η**do**  n←i+1  **for**
i=1→T
**do**   **for**
$j=1\to 2^{B}$
**do**    ${\left[\Delta \right]}_{j}\leftarrow {∥{\left[B_{n-1}\right]}_{:,j}-{\left[T\right]}_{:,i}∥}^{2}$   **end for**   $j_{sel}\leftarrow \underset{j}{\arg\min}\left|{\left[\Delta \right]}_{j}\right|$   $G_{j_{sel}}\leftarrow \left[\begin{array}{cc} G_{j_{sel}} & {\left[T\right]}_{:,i} \end{array}\right]$  **end for**  **for**
$i=1\to 2^{B}$
**do**   ${\left[B_{n}\right]}_{:,i}\leftarrow mean\left(G_{i}\right)$   ${\left[B\right]}_{:,i}\leftarrow {\left[B\right]}_{:,i}/∥{\left[B\right]}_{:,i}∥$  **end for**  $\mathrm{diff}\leftarrow {∥B_{n}-B_{n-1}∥}^{2}$  $G_{1},G_{2}\cdots ,G_{2^{B}}\leftarrow \left[\right]$**end while**$B\leftarrow B_{n}$Output:B

## 6. Simulation Results

This section shows simulation results for the NMSE of the recovered channels and a sum rate of multi-user downlink transmission. In the results, the proposed quantized channel feedback (QCF) scheme is compared with the linear quantized feedback scheme and the beam-steering method based on perfect CSIT. Parameters for simulation environment are presented in Table 1. To reflect the sparsity of the angle domain, the number of multi-path and scatterers are set to small numbers. The sparsity of the angle domain is shown in an environment that the number of propagation paths is close to one [3,28]. Furthermore, azimuth spread of departure (ASD), elevation spread of departure (ESD), azimuth spread of arrival (ASA), and elevation spread of arrival (ESA) are set to 3 degrees for the sparsity. For NMSE, the normalized error between the perfect channels and recovered channels at a transmitter is calculated as follows:(33)$$\mathrm{NMSE}\;\mathrm{of}\;{\hat{h}}_{u}=10\log\left(\frac{{∥h_{u}-{\hat{h}}_{u}∥}^{2}}{{∥h_{u}∥}^{2}}\right),$$
where $h_{u}$ and ${\hat{h}}_{u}$ are the perfect channel and the recovered channel at the transmitter. In the simulation for a sum rate, the number of receivers is set to four and the number of RF chains are set to the same number of receivers except for one case.

In Figure 3, NMSE measurements according to codebook size are presented. QCF-L means a QCF scheme using the linear codebook that is generated by the LBG algorithm in complex unit vector space. For feedback, the two schemes use the same linear codebook. The results of Figure 3 indicate that channel recovery using the common basis achieves the same NMSE with channel recovery using the selective basis. Therefore, there is no penalty to use the common sparse basis for basis of all OFDM subcarriers. When beamforming is used at the receivers, the same results are observed.

Figure 4 and Figure 5 also show NMSE measurements according to codebook size. In Figure 4 and Figure 5, QCF schemes using the linear codebook and the proposed codebook are compared. QCF-NL means the QCF scheme using the proposed nonlinear codebook. In the figures, the schemes use the same common basis for channel recovery. Common observation of Figure 4 and Figure 5 is that the nonlinear codebook achieves better performance than that of the linear codebook with the same codebook size. Specifically, systems can reduce 1.5∼2 bits per subcarrier to achieve the same performance with the non-linear codebook. Furthermore, the performance gap of the two codebooks decreases with the increase of the codebook size. In the case of $N_{t}=64$, the gap is smaller than the gap of $N_{t}=32$.

Figure 6 and Figure 7 show NMSE measurements according to SNR. For QCF schemes, a codebook of 6 bits is used. From the results, the nonlinear codebook shows better performance than the linear codebook and the performance gap is larger with the beamforming of the receivers. In the interval of 0 ∼ 5 dB SNR, the QCF schemes experience a lower effect of the noise than CS-OMP schemes. It seems that the basis selection of MMSE-OMP provides robustness to the noise even at the channel recovery of the transmitter.

Figure 8 and Figure 9 show measurements of a sum rate according to SNR. For the measurements, the number of receivers is set to four and the same SNR is assumed for all receivers. The beam-steering method uses the same number of RF chains with the number of receivers and the beams that are directed to the true elevation of departure (EOD) and the true azimuth of departure (AOD) for each receiver. For digital beamforming, ZF beamformer is used in all measurements. Furthermore, a fully connected structure is used for all hybrid beamforming. For the hybrid beamforming of the QCF schemes, the method of [9] is used to optimize jointly the analog and digital beamformers. As the proposed systems deliver compressed versions of the estimated channel matrices instead of indices for precoding matrices, a transmitter can adopt any beamforming schemes that use the channel matrix. In the results, the nonlinear codebook achieves a better sum rate than that of the linear codebook. The gap of the performance increases with SNR and is constant in the interval that the additional improvement for MNSE is not observed. The similar results are shown in the case of $N_{t}=64$. However, the slightly larger gap is observed at a low SNR environment than that of the case of $N_{t}=32$.

Figure 10 shows measurements of the case that the number of RF chains is twice the number of the receivers. For the beam-steering method, the same number of RF chains is used with the number of the receivers as the joint optimization cannot be performed. When the number of RF chains is twice the number of receivers, hybrid beamforming can achieve optimal beamforming gain with the joint optimization [8]. Although the QCF schemes use imperfect CSI, the schemes show a higher sum rate than that of the beam-steering method based on perfect CSIT. However, the improvement of the sum rate decreases drastically after the SNR of 10 dB. Furthermore, the improvement by the Rx beamforming decreases with the increase of SNR. It is can be thought that the sum rate is limited by imperfect suppression of the IUI. Therefore, the limited sum rate can be improved by the more accurate estimation of sparse vectors.

## 7. Conclusions

This paper presents a channel feedback scheme for multi-user hybrid beamforming systems to recover high-dimensional RF channel matrices effectively. The proposed scheme exploits the common sparsity of mmWave broadband channel and the property of OMP operation for codebook generation to reduce feedback overhead and improve the quality of the feedback information. From the results of NMSE for the recovered channels, it is shown that the scheme using the proposed codebook provides more accurate recovery with the same size of feedback than that of the channel recovery scheme with the linear codebook. Furthermore, the proposed feedback scheme allows a transmitter to adopt joint optimization for the hybrid beamformer. By the joint optimization, the systems can control the beamforming gain and the IUI flexibly. The NMSE of the recovered channel can be improved by using a more accurate estimation scheme for the sparse vectors than the simple OMP scheme. 

## Figures and Tables

**Figure 1 sensors-21-05298-f001:**
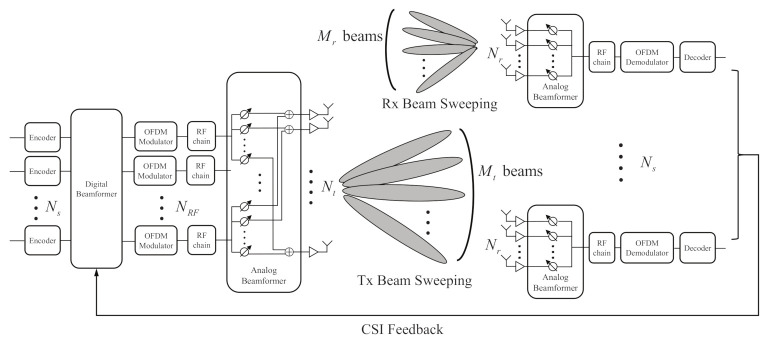
Multi-user hybrid beamforming system.

**Figure 2 sensors-21-05298-f002:**
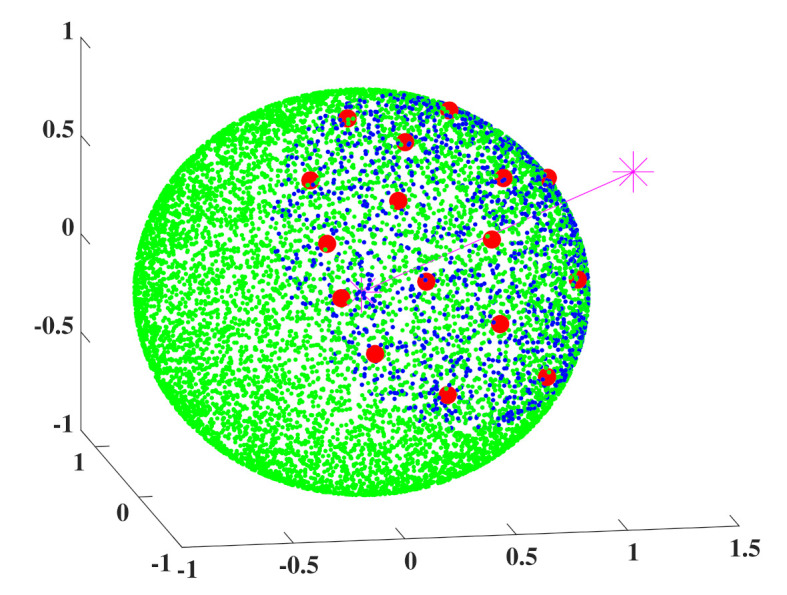
Codebook for conditional RVQ generated by the LBG algorithm.

**Figure 3 sensors-21-05298-f003:**
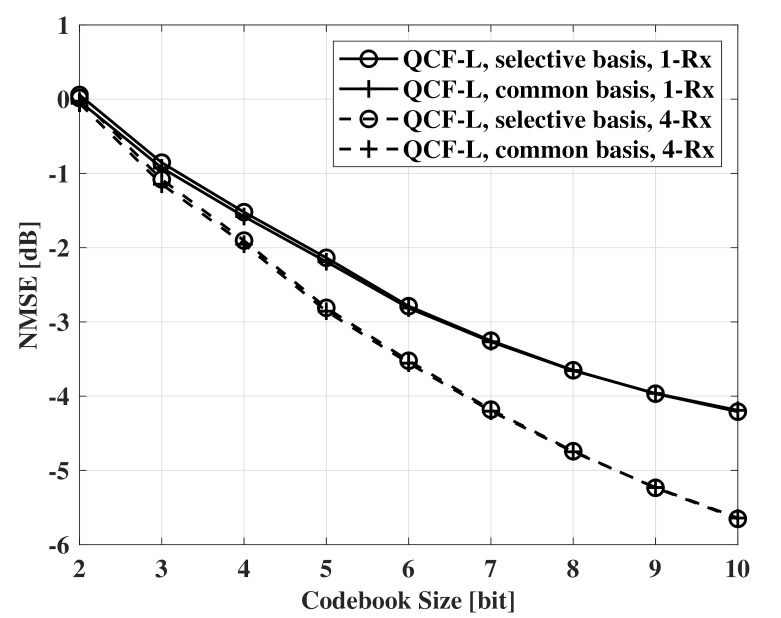
NMSE measurements according to codebook size, SNR=10dB, $N_{t}=32$, and $N_{r}=1,4$.

**Figure 4 sensors-21-05298-f004:**
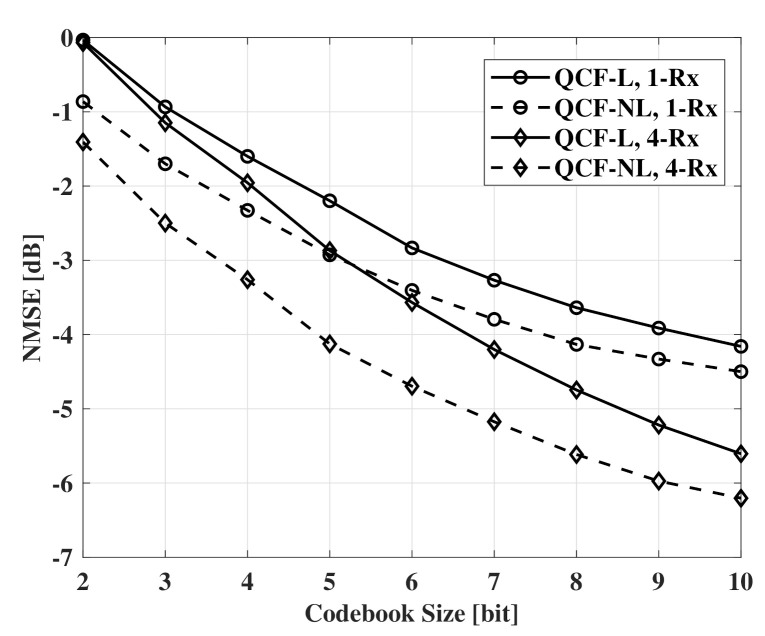
NMSE measurements according to codebook size, SNR=10dB, $N_{t}=32$, and $N_{r}=1,4$.

**Figure 5 sensors-21-05298-f005:**
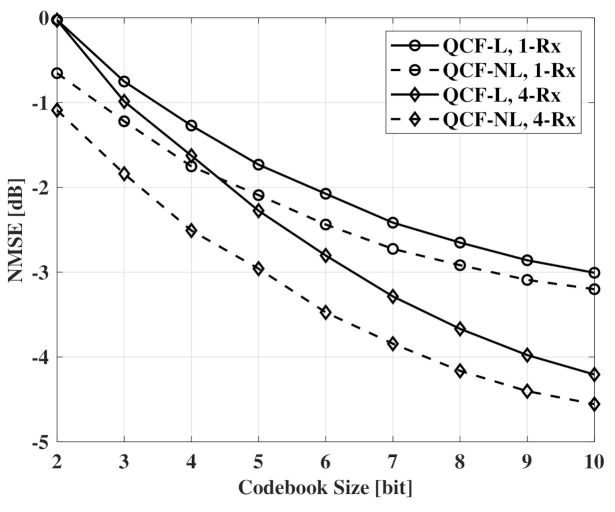
NMSE measurements according to codebook size, SNR=10dB, $N_{t}=64$, and $N_{r}=1,4$.

**Figure 6 sensors-21-05298-f006:**
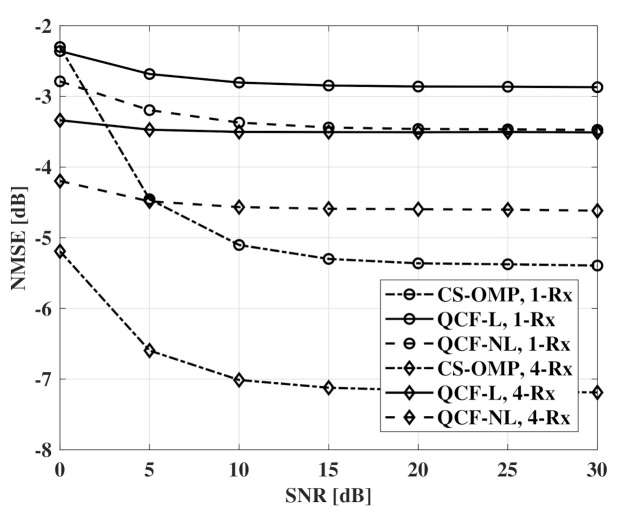
NMSE measurements according to SNR, codebooksize=6bits, $N_{t}=32$, and $N_{r}=1,4$.

**Figure 7 sensors-21-05298-f007:**
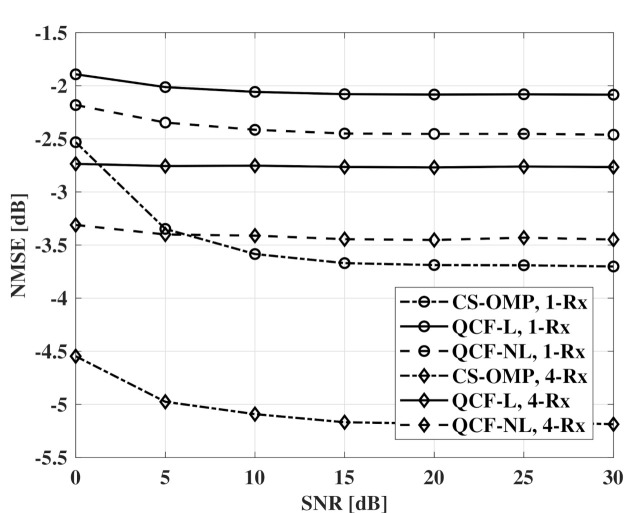
NMSE measurements according to SNR, codebooksize=6bits, $N_{t}=64$, and $N_{r}=1,4$.

**Figure 8 sensors-21-05298-f008:**
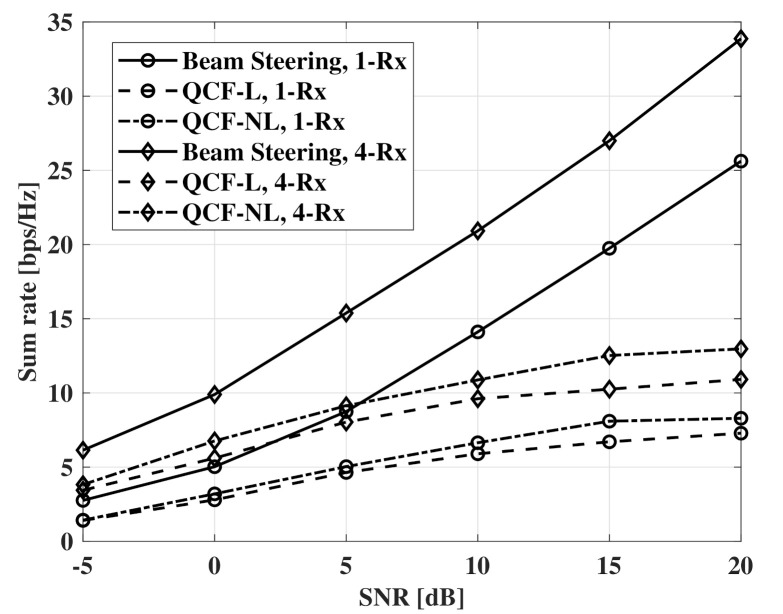
Sum rate measurements according to SNR, codebooksize=8bits, $N_{t}=32$, $N_{RF}=4$, and $N_{r}=1,4$.

**Figure 9 sensors-21-05298-f009:**
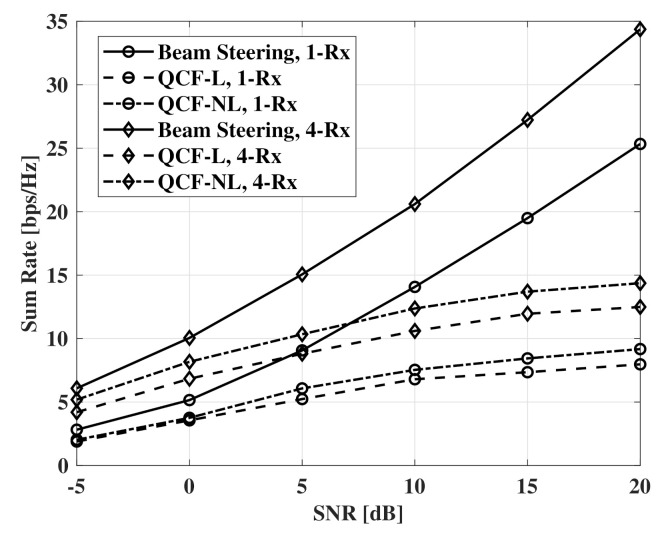
Sum rate measurements according to SNR, codebooksize=8bits, $N_{t}=64$, $N_{RF}=4$, and $N_{r}=1,4$.

**Figure 10 sensors-21-05298-f010:**
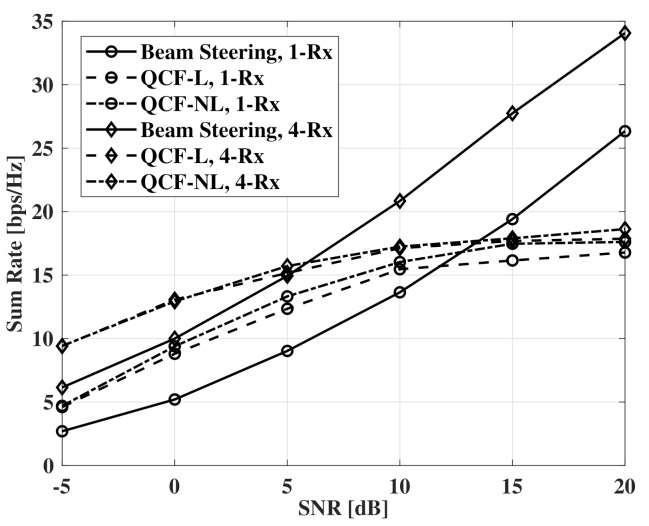
Sum rate measurements according to SNR, codebooksize=8bits, $N_{t}=64$, $N_{RF}=8,4$, and $N_{r}=1,4$.

**Table 1 sensors-21-05298-t001:** Simulation Parameters and Schemes.

Parameters	Value
Center frequency	30 GHz
Number of OFDM subcarriers	1024
Numeber of Tx antennas	64, 32
Number of RF chains	8, 4
Number of receivers	4
Number of Rx antennas	4, 1
Multi-path (non-line of sight)	3
Number of scatterers	10 per path
ASD, ESD, ASA, ESA	3 degree
Size of codebook	8, 6 bits
Compared scheme for quantization of sparse vectors	Linear RVQ
Compared schemes for a sum rate	Beam-steering, Method of [9]

## Data Availability

Not applicable.

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
