# Peer review of "Efficient Channel Feedback Scheme for Multi-User MIMO Hybrid Beamforming Systems"

_sensors, 2021, doi:10.3390/s21165298_

Round 1

Reviewer 1 Report

This is an interesting and well-written paper on channel feedback in multi-user MIMO systems with hybrid A/D beamforming. The authors capitalize their approach on the common sparsity of the wireless channels for all users and leverage on non-linear quantization; their proposed approach uses the minimum mean squared error orthogonal matching pursuit algorithm. The following need to be clarified:

1) What is the relevance of their approach to techniques based on pure beam searching, eg:

[R1] https://ieeexplore.ieee.org/document/9417259

[R2] https://ieeexplore.ieee.org/document/8313072

[R3] https://ieeexplore.ieee.org/document/8962355

2) The authors should discuss recent approaches on mmWave channel estimation exploiting the sparsity of the channels, eg:

[R4] https://ieeexplore.ieee.org/document/8812960

[R5] https://ieeexplore.ieee.org/document/7037320

Reviewer 2 Report

The introduction could be improved, mainly the paragraphs on page two. In line 78 which property is in mind? The channel reciprocity?

Equations (12) and (13) are not obvious. Could the authors give more details about how they reach there?

It is not clear if the sparsity of the mmwave channel in the angle domain stands for any channel conditions in this frequency range (see page 6) and for any number of antennas.

qβ in eq. (19) is also a vector for the beam sweeping as qn and pn in eqs. (2) and (3)?

The inclusion of a reference about OMP could be useful since the formulation for OMP is not clear.

Which kind of quantization is used in the non-linear codebook generation?

Why the multi-path is limited to 3? What happens for higher values?

The text on the paragraph after figure 3 must be improved.

Also, a space between the text and the square parenthesis is missing is the most references.

Round 2

Reviewer 2 Report

In this revision the authors have addressed the concerns from previous revision.